# A Dual-Branch Feature Fusion Framework based on GenAI for Robust Detection of Medical CT Image Tampering

## Abstract

CT images are a critical diagnostic tool in modern medicine, yet they face risks to image authenticity posed by diverse image tampering techniques, which could disrupt the normal medical order and the societal trust system. Although image tamper detection technology has made some progress, techniques specifically targeting CT image tampering detection are extremely scarce. This paper proposes a dual-branch feature fusion framework based on generative artificial intelligence (GenAI) for CT image tampering detection. This framework utilizes a ResNet-based generator to create tampered images that rich in edge and noise features, which are then fed into a dual-branch discriminator to separately learn noise and edge features. For the features learned by the discriminator, we designed a feature fusion module that captures complex relationships between features and selects different feature weights through a cross self-attention mechanism and dynamic feature selection. Additionally, we created a CT image tampering dataset and conducted comparative experiments with existing mainstream methods on public image tampering datasets and the self-made CT tampering image dataset. Experimental results demonstrate that the proposed method possesses good accuracy and robustness, providing an effective solution for CT image tamper detection.

## 1 Introduction

In modern medicine, computed tomography (CT) images have become one of the core tools for doctors in diagnosis and treatment decision-making, as noted by Khatami et al. (2017). However, with the rapid development of image tampering techniques, CT images face increasing risks of tampering, as reported by Pasqualino et al. (2024). Therefore, research on detection technologies for CT image tampering holds significant value and importance, as emphasized by Ghoneim et al. (2018). Traditional methods for image tamper detection are mainly categorized into detection based on tampering trace detection and intrinsic image feature statistical analysis, as described by Chen Yi Lei (2011). Common methods in tampering trace detection include block matching and feature vector analysis, resampling trace detection, and edge discontinuity detection Niu et al. (2021). Block matching involves dividing the image into fixed-size blocks, extracting feature vectors such as DCT coefficients, and identifying abnormal or repeated blocks to find tampered areas, particularly effective in copy-move operations Zhuang et al. (2021). Resampling trace detection uses periodic signals introduced by interpolation during operations to judge if scaling or rotation has occurred. Edge discontinuity detection locates anomalies by discovering inconsistencies in image edges after tampering Lin et al. (2011). Intrinsic image feature statistical analysis is achieved through pattern noise analysis and color channel correlation detection, where the former locates tampered areas by identifying regions lacking noise specific to imaging devices, and the latter detects tampering traces by utilizing changes in color channel correlation Barad & Goswami (2020). Additionally, deep learning technology has made remarkable strides in the field of image tamper detection, fundamentally transforming the way such tasks are approached Fridrich et al. (2003). Convolutional Neural Networks (CNN) are widely employed in image tamper detection due to their robust feature extraction capabilities Liu et al. (2022). CNN utilize a series of convolutional layers to progressively detect hierarchical patterns in the data, from simple edges and textures to more complex structures and tampering artifacts. By applying filters across an image, CNN can capture spatial hierarchies and relationships, enabling them

to identify subtle changes and anomalies indicative of tampering. This layered approach not only enhances the accuracy of detection but also allows for the efficient processing of high-dimensional image data. Moreover, CNN incorporate pooling operations that reduce the spatial dimensions of feature maps, which helps in managing computational complexity and preventing overfitting. Pooling layers, such as max pooling, condense information by selecting the most prominent features, aiding robustness against variations and distortions in the input data. This is particularly important in tamper detection, where the model must discern between genuine content and alterations that may be minuscule or concealed.

In recent years, the rapid development of Generative Adversarial Networks (GAN) has been demonstrated by Remya Revi et al. (2021), along with generative algorithms and pre-trained models, significantly improving the quality and diversity of Artificial Intelligence Generated Content (AIGC) Xu et al. (2024). This adversarial training between generation and discrimination greatly enhances the realism of model-generated images, while also promoting the learning of image details by the discriminator. The application of GAN in image tamper detection is also gradually increasing, as noted by Zhang et al. (2024), with researchers beginning to use its generative capabilities to enrich the training data of detection models while improving the ability to recognize complex tampering techniques. Traditional CNN detection methods face challenges when dealing with highly realistic generated images, and the application of GAN provides a new path to solve this problem. By deeply analyzing the characteristics of GAN-generated images, researchers have developed various detection techniques: on one hand using fake samples generated by GAN to enhance the diversity of the detection training set, and on the other hand combining GAN with deep learning techniques such as attention mechanisms and feature extraction technologies to further improve sensitivity to different types of tampering Xu et al. (2024). These methods perform excellently in capturing subtle and complex tampering traces, but still need to address the problem of high requirements for high-quality and diverse data. With further research, the combination of GAN and other detection technologies will help improve the accuracy and robustness of image tamper detection. Although image tamper detection methods have achieved certain detection effects in some areas, they still show inadequacies when addressing the specific challenge of CT image tampering. For CT image tamper detection, current methods mainly include machine learning algorithms Ghoneim et al. (2018), image feature analysis, and data integrity verification methods Chiang et al. (2008). However, these methods face technical limitations when dealing with complex tampering scenarios, including insufficient ability to recognize different types of tampering, strong dependency on specific datasets, and adaptability issues when addressing diverse CT image tampering techniques. Therefore, further in-depth research is needed to enhance the accuracy and robustness of these methods to effectively address diverse image tampering threats.

In this study, we aim to address the critical challenges associated with CT image tampering detection by proposing a novel dual-branch feature fusion framework based on generative artificial intelligence (GenAI). Existing methods, including traditional handcrafted feature-based approaches and modern transformer-based algorithms, exhibit limitations in handling the unique characteristics of medical images. Traditional methods often struggle with complex manipulations and fail to generalize across diverse tampering scenarios, while transformer-based approaches tend to emphasize global dependencies, potentially overlooking subtle local anomalies and fine-grained features essential for CT image analysis. To overcome these challenges, our framework integrates edge and noise features through a cross-attention mechanism, enabling precise localization of tampered regions. Furthermore, by leveraging the capabilities of GAN to produce realistic tampered images during training, our approach enhances robustness and generalization performance, making it well-suited for the complex and subtle nature of medical image tampering detection.

Our contributions are summarized as follows:

- We propose a GenAI-based dual-branch feature fusion framework, generating more tampered images with edge and noise features using a ResNet-based generator, which are fed into a dual-branch discriminator to learn noise and edge features in generated CT tampered images.

- Leveraging GenAI characteristics, we design a feature fusion module for the dual-branch discriminator, which includes a cross self-attention mechanism, residual connections, and dynamic feature selection. The cross self-attention mechanism captures complex relationships between features, residual connections alleviate gradient vanishing in deep networks,

and dynamic feature selection determines the weights of different features, improving the specificity of fused features.

- We create a CT image tampering dataset and conducted comparative experiments with existing mainstream methods on public image tampering datasets and a self-made CT tampered image dataset. Experimental results prove that our method has excellent accuracy and robustness.

## 2 RELATED WORK

### 2.1 MEDICAL IMAGE DETECTION

In recent years, deep learning has revolutionized medical image analysis, significantly improving tasks such as disease diagnosis, lesion localization, and organ segmentation Chen et al. (2017; 2023a). GAN-based methods, such as DH-GAN proposed by Liu et al. (2024), have shown promise in localizing manipulated regions by leveraging adversarial learning to enhance feature representation. The end-to-end Coupled GAN for multimodal feature fusion in Alzheimer's disease classification was introduced by Ma et al. (2021). In cancer diagnosis and lesion detection, Pasqualino et al. (2024) presented MITS-GAN to protect image authenticity against tampering attacks, while Tang et al. (2023) designed a dual-stream attention network based on ResNet variants to focus on texture and shape features for thyroid nodule diagnosis. For MRI, CNN and MIL-based methods Chen et al. (2023b); Farooq et al. (2017) have been extended by Lu et al. (2025) with a dual-branch framework using cross-attention loss to capture both spatial and local features. In multi-organ segmentation, Wan et al. (2024) proposed a single encoder multi-decoder design to construct and aggregate dependencies between organs, while semi-supervised approaches Chen et al. (2023c) improve encoder-decoder features via attention-guided predictors and semantic contrast learning to enhance performance when handling unlabeled data. Despite these advances, challenges remain in high-dimensional image analysis, subtle anomaly detection, and generalization across imaging modalities.

### 2.2 IMAGE TAMPERING DETECTION

Image tampering detection has become a major research area due to the increasing ease of manipulation with advanced editing tools Mehrjardi et al. (2023). A blocking strategy with rich model CNNs to process image blocks for robust localization was proposed by Zhou et al. (2017), while Yang et al. (2020) introduced Constrained R-CNN, a coarse-to-fine architecture learning unified manipulation features. Han et al. (2024) developed HDF-Net with RGB and SRM dual streams, integrating multiple saliency modules to improve accuracy, and Dong et al. (2022) designed a multi-view feature learning network exploiting boundary artifacts and noise cues. Liu et al. (2022) proposed PSCC-Net with spatial-channel correlation modules for multi-scale manipulation detection, and Huang et al. (2022) introduced DS-UNet to coarsely locate and refine tampering traces by revealing noise inconsistencies. Islam et al. (2020) applied a dual-attention GAN to capture positional and discriminative features for copy-move detection, while Remya Revi et al. (2021) explored both active and passive approaches, highlighting passive techniques such as AT-MobileViT and CSA blocks for AIGC image detection. These methods have shown strong results, but detecting subtle manipulations in complex domains remains challenging.

### 2.3 MEDICAL IMAGE TAMPERING DETECTION

Compared with general tampering detection, medical image tampering detection is less mature but has shown promising progress. Active protection methods Lu et al. (2024); Qasim et al. (2018), such as encryption and watermarking, aim to enhance the security and integrity of image data during transmission and storage. For example, a hybrid watermarking algorithm combining DWT and MobileNetV2 to improve robustness, concealment, and recovery capability under attacks was proposed by Nawaz et al. (2024). Passive detection methods are also gaining attention: Lin et al. (2023) introduced EMT-Net to enhance edge artifacts and capture subtle traces, and Bai et al. (2025) developed PIM-Net to exploit pixel- and region-level inconsistencies for fine localization. GAN-based frameworks such as MITS-GAN Pasqualino et al. (2024) add perturbations to improve anti-tampering capability in CT images, while Chiang et al. (2008) used wavelet transforms for detecting

and repairing tampered areas without original-image comparison. Ghoneim et al. (2018) combined multi-resolution regression filtering with machine learning for real-time tamper detection in cloud environments. However, many existing methods target specific image types or tampering scenarios, limiting generalizability. The complexity of anatomical structures and the requirement for high sensitivity in detecting subtle modifications motivate the development of more robust and adaptable frameworks, such as the one proposed in this work.

# 3 METHOD

## 3.1 OVERALL FRAMEWORK

Our proposed framework aims to effectively detect tampering in medical CT images, as shown in Figure 1. The generator in our framework plays a critical role in enhancing the model's generalization performance. Specifically, the generator takes random noise vectors as input, enabling it to introduce stochastic variations into the generated images, such as differences in edge sharpness, noise intensity, and boundary irregularities. Furthermore, the adversarial training process encourages the generator to continuously improve the realism of the tampered images, further contributing to the model's ability to generalize across different types of tampering scenarios. The discriminator is designed with a dual-branch structure, learning edge and noise features separately. The edge discriminator branch (D1) extracts edge information using edge enhancement techniques and a U-Net edge segmentation module. This branch aids in detecting boundary discontinuities in images, especially important for detecting tampering involving geometric adjustments or morphological changes. By combining U-Net's feature extraction capabilities with edge enhancement mechanisms, this branch effectively detects anomalies in edge distribution. The noise discriminator branch (D2) enhances subtle and critical noise features in the image using Spatial Rich Model (SRM) filters. In medical images, slight noise changes may indicate tampering traces. By applying SRM filters, this branch identifies noise differences produced during tampering, detecting tampering traces in CT images.

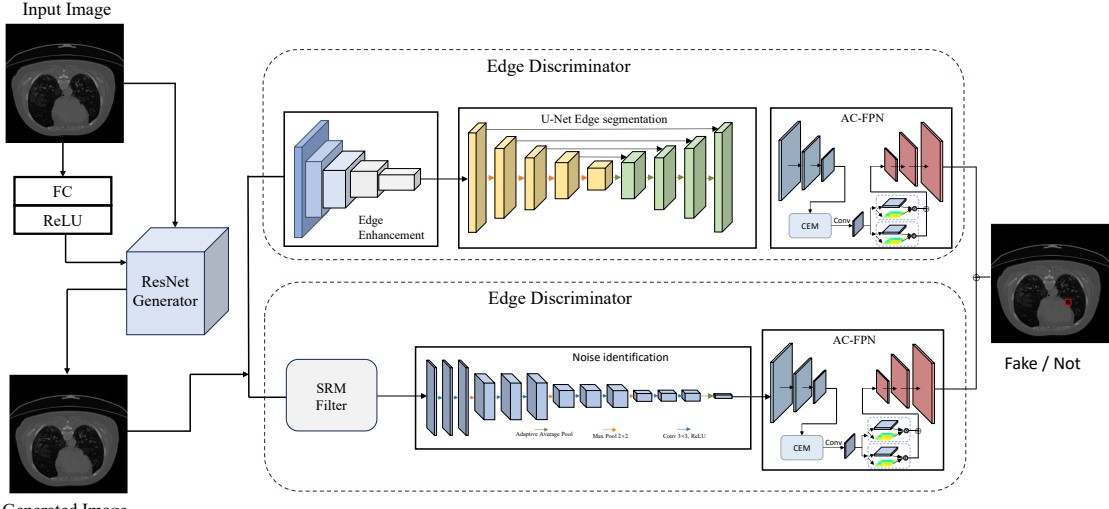

Figure 1: Proposed Network Structure

## 3.2 EDGE DISCRIMINATOR BRANCH

The edge discriminator branch ($D1$) focuses on extracting edge features from the generated CT images, which are crucial for identifying abnormal boundaries caused by tampering.

First, the edge enhancement module applies a Sobel filter to identify and amplify edge features in the image:

$$E_{\text{enhanced}} = \sqrt{(G_x * I_{\text{input}})^2 + (G_y * I_{\text{input}})^2} \tag{1}$$

where $G_x$ and $G_y$ represent the convolution kernels of the Sobel filter in the horizontal and vertical directions, and $*$ denotes the convolution operation.

Next, our U-Net variant processes the edge-enhanced feature map, combining encoder, decoder, and skip connections to extract hierarchical and multi-scale edge features, which enhances sensitivity to tampering details:

$$F_{\text{seg}} = \text{U-Net}(E_{\text{enhanced}}) \tag{2}$$

After U-Net completes the segmentation, the AC-FPN (Adaptive Contextual Feature Pyramid Network) module performs multi-scale fusion on the segmented edge features. AC-FPN optimizes edge feature expression by integrating features at different scales. First, the input features are decomposed into representations at different scales through multi-scale feature extraction:

$$F_{\text{pyramid}}^{i} = \text{Downsample}(F_{\text{seg}}), \quad i = 1, 2, \ldots, m \tag{3}$$

where $F_{\text{fused}}$ represents the final fused feature map after multi-scale integration, and $F_{\text{pyramid}}^{i}$ denotes the feature map at scale $i$ extracted by the multi-scale feature extractor. $\beta_i$ are learnable weights that determine the contribution of each scale feature map during fusion. Upsample($\cdot$) refers to the upsampling operation, which restores the spatial resolution of the feature maps to match the original input size. This weighted summation ensures that features across different scales are effectively combined to enhance representation.

Then, features at different scales are integrated through weighted summation:

$$F_{\text{fused}} = \sum_{i=1}^{m} \beta_i \cdot \text{Upsample}(F_{\text{pyramid}}^{i}) \tag{4}$$

where $\beta_i$ are learned weights used to control the contribution of features at different scales.

Finally, through the AC-FPN module, features are integrated to extracte more detailed feature expressions.

### 3.3 Noise Discriminator Branch

The noise discriminator branch (D2) is designed to enhance and extract noise-related features, which are critical for detecting tampering artifacts in CT images. First, Spatial Rich Model (SRM) filters and the wavelet transform $\mathcal{W}(\cdot)$ are combined to decompose the input image $I_{\text{input}}$ into multi-frequency representations:

$$W_i = \mathcal{W}(I_{\text{input}}) * H_i \tag{5}$$

where $H_i$ denotes a set of high-pass SRM filters that enhance noise features in different directions. Specifically, three representative SRM filters are used:

$$H_1 = \begin{bmatrix} -1 & 2 & -1 \\ 2 & -4 & 2 \\ -1 & 2 & -1 \end{bmatrix}, \quad H_2 = \begin{bmatrix} 0 & 0 & 0 \\ 1 & -2 & 1 \\ 0 & 0 & 0 \end{bmatrix}, \quad H_3 = \begin{bmatrix} -1 & 2 & -1 \\ 0 & 0 & 0 \\ 1 & -2 & 1 \end{bmatrix} \tag{6}$$

These filters help highlight noise inconsistencies caused by tampering. After this preprocessing, the feature maps are further refined using a series of convolutional operations to form detailed noise representations.

For enhanced discrimination, an Adaptive Contextual Feature Pyramid Network (AC-FPN) module is employed to aggregate features at multiple scales. Down-sampling and up-sampling techniques are combined for effective multi-scale fusion:

$$F_{\text{fused}} = \sum_{i=1}^{n} \alpha_i \cdot F_i \tag{7}$$

where $F_i$ denotes the feature at scale $i$ and $\alpha_i$ are dynamically learned weights that adaptively emphasize the most tampering-sensitive features.

To further guide the network with anatomical context, an anatomical prior map $A_{\text{prior}}$ is introduced. An attention mechanism $\mathcal{A}(\cdot)$ calculates attention weights $M_{\text{attn}}$ for improved feature calibration:

$$M_{\text{attn}} = \mathcal{A}(F_{\text{res}}, A_{\text{prior}}), \quad F_{\text{adj}} = M_{\text{attn}} \odot F_{\text{res}} \tag{8}$$

where $\odot$ denotes element-wise multiplication.

Finally, the adjusted noise feature map $F_{\text{adj}}$ is flattened and passed through a fully connected layer for final decision:

$$F_{\text{final}} = \text{FC}(\text{Flatten}(F_{\text{adj}})) \tag{9}$$

where $\text{Flatten}(\cdot)$ operation converts the feature map into vector form for processing by the fully connected layer. This process ensures that the final expression of noise features can effectively support tamper detection decisions.

## 3.4 AC-FPN MODULE (PARTIAL)

In our dual-branch framework, the AC-FPN and feature fusion modules are critical for enhancing detection sensitivity and accuracy. The Adaptive Contextual Feature Pyramid Network (AC-FPN), shown in Figure 2, builds on the traditional Feature Pyramid Network (FPN) framework, but is adapted for the unique characteristics of medical images.

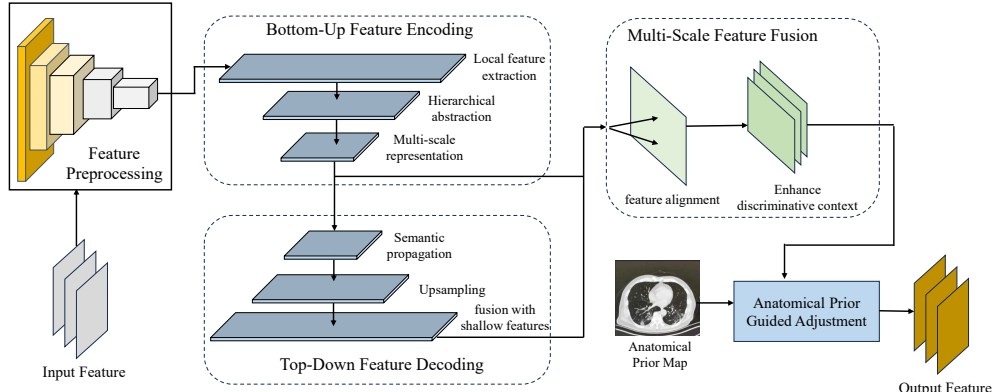

Figure 2: Detailed architecture of the AC-FPN module.

The AC-FPN first receives input features from the generator, which are processed through a series of convolutional layers:

$$F_{\text{conv}}^i = \sigma(W^i * F_{\text{input}} + b^i), \tag{10}$$

where $F_{\text{conv}}^i$ denotes the output of the $i$-th convolutional layer, $*$ the convolution operation, and $\sigma$ an activation function (typically ReLU).

Multi-scale processing follows a bottom-up path, where features from all layers are aggregated:

$$F_{\text{bottom-up}} = \sum_{j=1}^{n} \alpha_j \cdot \text{Pool}(F_{\text{conv}}^j), \tag{11}$$

with $\alpha_j$ as learnable weights and $\text{Pool}(\cdot)$ representing a pooling operation.

## 3.5 FEATURE FUSION MODULE

Attention maps for each branch are first computed using learnable weight matrices:

$$A_{D1} = \text{softmax}(W_{D1} F_{D1}), \quad A_{D2} = \text{softmax}(W_{D2} F_{D2}), \tag{12}$$

The attention maps are then integrated by element-wise multiplication:

$$A_{combined} = A_{D1} \odot A_{D2}, \tag{13}$$

The combined attention map is used to weight the sum of feature maps from both branches, yielding the final fused feature:

$$F_{final} = A_{combined} \odot (F_{D1} + F_{D2}) \tag{14}$$

Furthermore, multi-scale fusion is performed by weighted summation:

$$F_{\text{acfpn}} = \sum_{i=1}^{m} \gamma_i \cdot F_{\text{scale}}^i, \tag{15}$$

where $F_{\text{scale}}^i$ are features at different scales and $\gamma_i$ are learned weights.

Guided by anatomical prior maps, this process ensures fused features can better reflect anatomical structures and tampered regions. During training, adversarial loss, edge loss, and noise loss are combined, with the total loss formulated as:

$$
\begin{aligned}
L_{total} &= \alpha \, L_{\text{adv}} + \beta \, L_{\text{edge}} + \gamma \, L_{\text{noise}} + \delta \, L_{\text{attn}} \\
&= \alpha \left( -\frac{1}{n_r} \sum (1 - \bar{p}(real))^2 \log D(x_r) \right) \\
&\quad + \beta \left( -\frac{1}{n_f} \sum (1 - \bar{p}(fake))^2 \log(1 - D(G(z))) \right) \\
&\quad + \gamma \frac{1}{n_r} \sum \|F_{D1}(x_r) - F_{D1}(G(z))\|_2^2 \\
&\quad + \delta \frac{1}{n_r} \sum \|F_{D2}(x_r) - F_{D2}(G(z))\|_2^2
\end{aligned}
\tag{16}
$$

## 4 EXPERIMENTS

### 4.1 EXPERIMENTAL PREPARATIONS

#### 4.1.1 PUBLIC AND CUSTOM CT TAMPERED IMAGE DATASETS

We evaluate our method on multiple public tampered image datasets, including CASIAv2, Columbia, NIST16, and MSMC series datasets, which contain diverse manipulation types such as copy-move, splicing, and inpainting. Detailed dataset composition, image counts, and manipulation type breakdowns are reported in Appendix A.4, along with Table 5.

In addition to these existing datasets, we created a new dataset specifically for CT image tampering detection. As shown in Figure 3, the workflow begins with using the SAM2 model Ravi et al. (2024) for region segmentation, which allows for accurate extraction of lesion areas based on coarse annotations from the original CT image dataset. By randomly copying and splicing lesion areas within the original CT images, the lesions are specified and extracted, then subjected to transformations such as shuffling, rotation, and scratch removal to enhance their diversity. These tampered lesions are then spliced back into the images within valid areas, ensuring realistic and effective tampering.

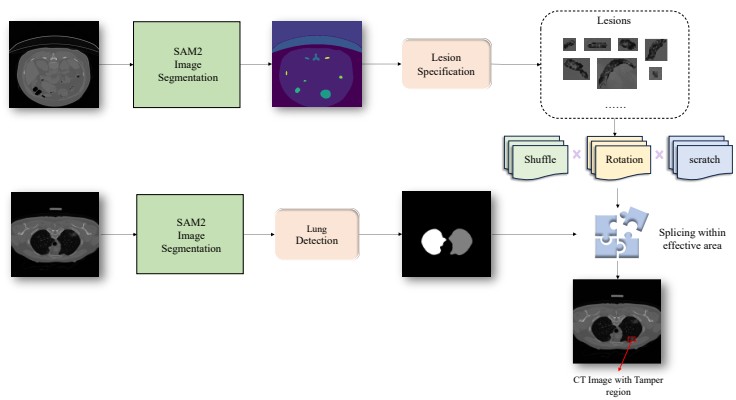

Figure 3: The workflow for creating the custom CT image dataset. This process involves the segmentation and manipulation of lesions to generate diverse tampered images.

Our custom dataset comprises a total of 6,000 images, equally divided into two types of tampering: 3,000 images feature copy-move tampering, while the remaining 3,000 images involve splicing tampering. This dataset is designed to provide a comprehensive dataset for evaluating the accuracy of tampering detection algorithms, specifically in the context of CT images.

## 4.2 COMPARISON ON PUBLIC TAMPERED IMAGE DATASETS

In the first experiment, we evaluated the performance of our model on public tampered image datasets. This experiment aimed to compare the detection capabilities of our model with those of existing state-of-the-art methods. We used the CASIAv2, Columbia, NIST16, and MSMC-16k datasets for this evaluation Dong et al. (2013); National Institute of Standards and Technology (2016); Ren et al. (2022). The results of this experiment are presented in Table 1.

| Methods | F1/AUC(%) | | | |
|---|---|---|---|---|
| | CasiaV1+ | Columbia | NIST | MSMC-16k |
| CR-CNNYang et al. (2020) | 34.5/42.0 | 25.0/52.0 | 20.0/48.0 | 21.5/47.0 |
| ManTra-NetWu et al. (2019) | 54.8/70.0 | 39.4/70.0 | 28.7/50.0 | 23.1/55.0 |
| HiFi-NetZheng et al. (2024) | 44.6/68.0 | 44.9/73.0 | 26.9/48.0 | 24.4/56.0 |
| CAT-NetKwon et al. (2021) | 48.1/60.0 | 46.2/78.0 | 36.1/63.0 | 37.7/65.0 |
| MVSSDong et al. (2022) | 72.4/80.0 | 42.8/71.0 | 39.3/76.0 | 39.5/70.0 |
| PSCCLiu et al. (2022) | 72.6/81.5 | 66.3/95.1 | 49.3/84.9 | 36.5/70.1 |
| Our model | **78.1/86.0** | **70.3/90.0** | **68.2/89.0** | **60.6/80.0** |

Table 1: Performance on various image tampering detection datasets

Specifically, on the MSMC-16k dataset, our model achieved an F1 score of 60.6% and an AUC of 80.0%. This result highlights the model's capability to effectively detect tampered regions in diverse manipulation scenarios, demonstrating its accuracy in handling complex and varied tampering techniques. On the CasiaV1+ dataset, our model reached an F1 score of 78.1% and an AUC of 86.0%, showcasing its high precision and sensitivity in identifying tampered areas. For the Columbia dataset, an F1 score of 70.3% and an AUC of 90.0% were achieved, indicating the model's effectiveness in minimizing false positives, especially in splicing tampered images. Lastly, in the NIST dataset, the model delivered an F1 score of 68.2 and an AUC of 89.0%, confirming its strength in detecting tampering in high-resolution and structurally complex images.

## 4.3 COMPARISON ON CT IMAGE DATASET

In this experiment, we assessed the performance of our model on our CT image dataset. This experiment was designed to evaluate the model's ability to detect tampering in CT images, which present different challenges compared to public image datasets. The results are shown in Table 2.

Our model demonstrated outstanding performance on the our CT image dataset, achieving the highest overall F1 score and AUC compared to other leading methods. For copy-move tampering, the model achieves an F1 score of 86.1 and an accuracy of 88.0, underscoring its ability in highlighting subtle inconsistencies introduced during the tampering process. This is largely due to the sophisticated noise feature extraction capabilities of the model, particularly the use of the SRM filter. In the case of splicing tampering, our

| Methods | F1/AUC(%) | | |
|---|---|---|---|
| | Copy-Move | Splicing | Overall |
| CR-CNN | 65.3 / 67.0 | 62.4 / 64.5 | 64.1 / 65.2 |
| ManTra-Net | 70.4 / 72.3 | 69.1 / 71.4 | 68.7 / 70.8 |
| HiFi-Net | 62.7 / 64.5 | 66.5 / 68.7 | 65.0 / 66.2 |
| CAT-Net | 71.8 / 73.6 | 68.5 / 70.9 | 70.6 / 72.1 |
| MVSS | 78.5 / 80.7 | 76.8 / 79.0 | 77.7 / 79.9 |
| PSCC | 80.2 / 82.5 | 78.9 / 81.1 | 79.6 / 81.8 |
| **Our model** | **86.1 / 88.0** | **83.7 / 85.6** | **85.9 / 87.4** |

Table 2: Performance on CT image tampering dataset

model reached an F1 score of 83.7% and an accuracy of 85.6%, demonstrating its ability to detect unnatural boundaries and inconsistencies. The overall performance, with an F1 score of 85.9% and an accuracy of 87.4%, reflects the model's comprehensive analysis capabilities, enabled by its dual-branch structure and feature fusion module.

## 4.4 ABLATION STUDY

In this ablation study, we systematically evaluated the contributions of individual components within our model's architecture to understand their impact on the overall performance of image tampering detection. This involved selectively disabling each branch of the dual-branch network and substituting the backbone network with ResNet-50 and ResNet-101 architectures. The results are presented in Table 3.

| Modified Structure | CasiaV1+ | Columbia | NIST | MSMC-16k | CT Dataset |
|---|---|---|---|---|---|
| *Discriminator Branch:* | | | | | |
| Noise Discriminator | 30.1 / 44.5 | 29.3 / 41.8 | 31.7 / 43.2 | 28.6 / 40.1 | 33.9 / 47.0 |
| Edge Discriminator | 38.2 / 49.6 | 36.5 / 45.9 | 39.8 / 48.3 | 37.1 / 46.7 | 40.5 / 50.2 |
| *Backbone Network:* | | | | | |
| ResNet-50 | 43.7 / 53.1 | 41.5 / 49.7 | 45.9 / 52.6 | 42.2 / 50.9 | 46.8 / 56.0 |
| ResNet-101 | 46.9 / 56.4 | 44.8 / 52.8 | 47.2 / 54.3 | 48.3 / 55.9 | 50.1 / 58.7 |
| *AC-FPN Module:* | | | | | |
| Without AC-FPN | 65.2 / 72.5 | 62.1 / 68.9 | 59.8 / 65.4 | 50.3 / 60.1 | 70.5 / 73.2 |
| Simple Upsampling | 68.4 / 75.3 | 65.7 / 72.1 | 62.5 / 69.8 | 54.6 / 63.9 | 75.2 / 78.5 |
| **Full Model (with AC-FPN)** | **78.1 / 86.0** | **70.3 / 90.0** | **68.2 / 89.0** | **60.6 / 80.0** | **85.9 / 87.4** |

Table 3: Performance of modified structures in ablation study.

From Table 3, the noise-branch classifier alone achieved 30.1%/44.5% F1/AUC on CasiaV1+, while the visual-branch scored 38.2%/49.6%, highlighting the need to capture both noise and edge features. Replacing the dual-branch architecture with single backbones like ResNet-50 or ResNet-101 led to notable performance drops, underscoring the value of the integrated design with cross-attention. The AC-FPN module also proved critical: removing it reduced CT dataset results to 70.5%/73.2%, and simple upsampling achieved only 75.2%/78.5%, both below the full model's 85.9%/87.4%.

## 4.5 ROBUSTNESS TEST

To evaluate our model's robustness under real-world attack scenarios, we conducted experiments involving common types of image perturbations. The detailed settings for these attacks are provided in Appendix A.7, and the results are shown in Table 4.

The model achieved an F1 score of 68.9% under rotational attacks, showcasing its robust noise feature extraction capabilities. For scaling disturbances, it achieved an F1 score of 67.4, supported by the edge feature extraction branch's ability to detect structural inconsistencies. Under noise attacks, the model maintained an F1 score of 66.8%, indicating its effectiveness in distinguishing genuine tampering signals from noise artifacts. Overall, the model's

| Methods | F1/AUC(%) under various attacks | | | |
|---|---|---|---|---|
| | Rotation | Scaling | Noise | Overall |
| CR-CNN | 45.2 / 47.0 | 44.1 / 46.0 | 43.5 / 45.3 | 44.3 / 46.1 |
| ManTra-Net | 50.3 / 52.2 | 48.7 / 50.5 | 47.9 / 49.8 | 49.0 / 50.8 |
| HiFi-Net | 44.8 / 46.5 | 46.2 / 48.0 | 45.5 / 47.3 | 45.5 / 47.3 |
| CAT-Net | 52.7 / 54.5 | 51.3 / 53.0 | 50.5 / 52.3 | 51.5 / 53.3 |
| MVSS | 58.0 / 60.1 | 56.8 / 58.9 | 55.7 / 57.8 | 56.8 / 58.9 |
| PSCC | 60.5 / 62.7 | 59.2 / 61.3 | 58.3 / 60.5 | 59.3 / 61.5 |
| **Our Model** | **68.9 / 70.8** | **67.4 / 69.3** | **66.8 / 68.7** | **67.7 / 69.6** |

Table 4: Robustness testing results under various attacks

performance, with an F1 score of 67.7% and an AUC of 69.6%, highlights its potential for ensuring medical image integrity, even in the presence of various attacks.

## 5 CONCLUSION

This paper presents an innovative dual-branch feature fusion framework based on GenAI for tamper detection in CT images. The framework combines GenAI with a dual-branch feature fusion module, capturing subtle tampering features in CT images—primarily edge and noise features—through a dual-branch discriminator. It achieves feature fusion via a cross-attention mechanism, ultimately enabling accurate localization of tampered areas in CT images. In experiments, our method demonstrated excellent performance across multiple public tampering datasets and a custom CT image tampering dataset. Additionally, we tested under various image attack conditions, and the results confirmed the robustness of our model against image disturbances. The proposed method holds significant importance in ensuring the authenticity of medical images and offers new perspectives and solutions for future research in medical image tamper detection.

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

## A  APPENDIX

### A.1  AC-FPN MODULE

To further capture information across different levels, aggregation and top-down fusion are performed:

$$F_{\text{output}} = \phi\big(\gamma \cdot \text{Upsample}(F_{\text{bottom-up}}) + \delta \cdot F_{\text{skip}}\big), \tag{17}$$

where $F_{\text{skip}}$ denotes skip connections, $\gamma$ and $\delta$ are balancing coefficients, Upsample$(\cdot)$ denotes upsampling, and $\phi(\cdot)$ is a refinement function such as additional convolution and non-linear activation.

This structure enables AC-FPN to integrate and refine features at multiple scales, improving the model's ability to capture both global context and fine details for robust tampering detection.

## A.2   IMPLEMENTATION DETAILS

In our dual-branch feature fusion framework, we implemented using the PyTorch deep learning framework and conducted training and evaluation on dual NVIDIA RTX 4090 GPUs. During training, the Adam optimization algorithm was used to adjust the parameters of the generator and discriminator, with an initial learning rate of $10^{-4}$, dynamically reduced to $10^{-8}$ using the ReduceLROnPlateau algorithm to promote optimal model convergence. Training was conducted using a mini-batch strategy with a batch size of 32, and the number of training epochs was set to 100. To prevent overfitting, a dropout rate of 0.5 was applied in the fully connected layers, and weight decay techniques were used. For the weight factors of the loss function, $\alpha$ and $\beta$ were set to 1 to balance the loss of the generation and discrimination process, while $\gamma$ and $\delta$ were set to 0.1 to fine-tune the impact of feature matching.

The overall training process of our framework is outlined in Algorithm 1. Each input medical CT image undergoes normalization preprocessing, and noise reduction techniques such as Gaussian filtering are applied to preserve basic details. The generator network generates synthetic images from random noise vectors to challenge the discriminator. The first discriminator (D1) combines edge enhancement techniques and U-Net structures to extract edge features, while the second discriminator (D2) uses Spatial Rich Model (SRM) filters to extract noise features. The features of the discriminators are integrated through a cross-attention mechanism to enhance feature representation, computing the total loss and guiding the optimization of the model. During testing, the model applies the same preprocessing and generation steps to each test image, generating tampering probability maps for authenticity judgment.

## A.3   ALGORITHM

Algorithm 1 outlines the complete training and testing procedure of our proposed dual-branch feature fusion framework for CT image tampering detection. During training, each input CT image undergoes normalization and Gaussian filtering to preserve important structural details, followed by edge enhancement using a Sobel filter. The generator $G$ produces synthetic tampered images from the preprocessed inputs and random noise vectors, introducing diverse edge and noise characteristics. The edge discriminator branch ($D1$) employs a U-Net variant and AC-FPN to extract and refine edge features, while the noise discriminator branch ($D2$) utilizes SRM filters, wavelet transform, and AC-FPN to capture noise-related tampering traces. A cross-attention mechanism integrates features from both branches, and the total loss $L_{total}$ combines adversarial, edge, noise, and cross-attention losses to update the generator and discriminators. During testing, the same preprocessing is applied, and the trained generator and discriminators produce edge and noise feature maps, which are fused to generate a tampering likelihood map $T(x, y)$, from which the authenticity verdict is derived.

**Input:** Input CT image $I$
**Output:** Authenticity assessment result
**for** *each training step from* 1 *to* $N$ **do**

   Preprocess image $I$: normalize, apply Gaussian filtering, and enhance edges using Sobel filter.

   Generate synthetic image $\widehat{I} = G(I, z)$ using generator $G$ with input image $I$ and random noise $z$.

   Apply edge enhancement and extract edge features $E_{\text{enhanced}}$ using Sobel filter.

   Process $E_{\text{enhanced}}$ through U-Net variant to get edge feature map $F_{\text{seg}}$.

   Perform multi-scale fusion using AC-FPN to obtain refined edge features.

   Compute edge loss $L_{edge}$ based on feature differences.

   Apply SRM filters and wavelet transform to extract noise features.

   Process through AC-FPN to obtain refined noise features.

   Compute noise loss $L_{noise}$ based on feature differences.

   Integrate features from D1 and D2 using cross-attention mechanism.

   Compute cross-attention loss $L_{CA}$.

   Compute total loss $L_{total} = \alpha L_{\text{adv}} + \beta L_{edge} + \gamma L_{noise} + \delta L_{CA}$.

   Update generator $G$ and discriminators $D1$, $D2$ using $L_{total}$.

**end**
**for** *each testing image* $I_{test}$ **do**

   Preprocess $I_{test}$: normalize, apply Gaussian filtering, and enhance edges.

   Generate synthetic image $\widehat{I_{test}} = G(I_{test}, z)$ using trained generator $G$.

   Compute edge feature map using edge discriminator $D1$ on $\widehat{I_{test}}$.

   Compute noise feature map using noise discriminator $D2$ on $\widehat{I_{test}}$.

   Combine edge and noise features to produce the final tampering likelihood map $T(x, y)$.

   Output the tampering likelihood map $T(x, y)$ and derive the final authenticity verdict.

**end**

**Algorithm 1:** Training and Testing Procedure

## A.4 PUBLIC DATASET DETAILS

Our training dataset primarily consists of CASIAv2 Dong et al. (2013) and MSMC-80k, while the testing dataset includes CasiaV1+, Columbia Ng et al. (2009), NIST National Institute of Standards and Technology (2016), and MSMC-16k, with dataset details shown in Table 5. To construct a robust and diverse dataset for validating our model, we created two unique datasets: MSMC-80k and MSMC-16k. The images in these datasets come from DEFACTO Mahfoudi et al. (2019), MSM30K Ren et al. (2022), and tampered images made using MS-COCO as the original images. MSMC-80k is our primary training dataset, containing a total of 80,000 images, including 17,739 copy-move images, 40,287 splicing images, and 21,974 inpainting images. On the other hand, MSMC-16k is designed as a testing dataset, containing 16,000 images, with 2,500 copy-move images, 2,500 splicing images, and 3,000 inpainting images. We adopt a strict sampling method to ensure that source images used in training do not appear in the testing set, thus preventing data leakage and maintaining the integrity of the evaluation.

| Dataset | Number | copy-move | splicing | impainting |
|---|---|---|---|---|
| *Training* | | | | |
| MSMC-80k | 80000 | 17739 | 40287 | 21974 |
| CASIAv2 | 5063 | 3235 | 1828 | 0 |
| *Testing* | | | | |
| CasiaV1+ | 200 | 100 | 0 | 0 |
| Columbia | 363 | 0 | 180 | 0 |
| NIST | 564 | 68 | 288 | 208 |
| MSMC-16k | 16000 | 2500 | 2500 | 3000 |

Table 5: The two training datasets and four testing datasets

## A.5 Our Tampered CT Image Dataset

In addition to the existing datasets, we created a new dataset specifically for CT image tampering detection. As shown in Figure 3, the workflow begins with using the SAM2 model Ravi et al. (2024) for region segmentation, which allows for accurate extraction of lesion areas based on coarse annotations from the original CT image dataset. By randomly copying and splicing lesion areas within the original CT images, the lesions are specified and extracted, then subjected to transformations such as shuffling, rotation, and scratch removal to enhance their diversity. These tampered lesions are then spliced back into the images within valid areas, ensuring realistic and effective tampering.

Our custom dataset comprises a total of 6,000 images, equally divided into two types of tampering: 3,000 images feature copy-move tampering, while the remaining 3,000 images involve splicing tampering. This dataset is designed to provide a comprehensive dataset for evaluating the accuracy of tampering detection algorithms, specifically in the context of CT images, which pose unique challenges due to their complex anatomical structures and small targets. While our study primarily focuses on copy-move and splicing tampering, which are the most common and impactful tampering methods in medical imaging scenarios, we acknowledge the existence of other tampering behaviors such as deletion and inpainting. These methods, often used to erase or replace specific regions in an image, are equally important to address in tampering detection. However, due to the lack of publicly available and suitable datasets for CT images containing these tampering types, they were not included in our current experiments.

## A.6 Evaluation Metrics

The F1 score is a harmonic mean of precision and recall, providing a balanced measure that considers both false positives and false negatives. It is defined as:

$$\text{F1 Score} = 2 \cdot \frac{\text{Precision} \cdot \text{Recall}}{\text{Precision} + \text{Recall}}, \tag{18}$$

where precision is the proportion of correctly identified tampered pixels (true positives, TP) to all pixels predicted as tampered (true positives + false positives, TP + FP), and recall is the proportion of correctly identified tampered pixels (true positives, TP) to all actual tampered pixels (true positives + false negatives, TP + FN). The F1 score is particularly useful in scenarios where there is an imbalance between tampered and non-tampered regions, as it ensures that both precision and recall are equally weighted. In our experiments, the F1 score was computed at both the pixel level and the image level, with a threshold of 0.5 for binary classification.

The Area Under the Curve (AUC) metric measures the area under the Receiver Operating Characteristic (ROC) curve, which plots the true positive rate (TPR) against the false positive rate (FPR) at various threshold settings. The AUC is defined as:

$$\text{AUC} = \int_0^1 \text{TPR}(t) \cdot d\text{FPR}(t), \tag{19}$$

where the true positive rate (TPR) is given by $\text{TPR} = \frac{\text{TP}}{\text{TP}+\text{FN}}$, and the false positive rate (FPR) is given by $\text{FPR} = \frac{\text{FP}}{\text{FP}+\text{TN}}$. A higher AUC value indicates better model performance in distinguishing between tampered and non-tampered regions. Unlike threshold-dependent metrics, the AUC evaluates the model's performance across various decision thresholds, making it a robust indicator of overall classification ability. In this study, the AUC was calculated for each dataset to evaluate the model's capability to generalize across different tampering scenarios.

## A.7 Robustness Test Details

In our robustness evaluation, three common types of attacks were applied to simulate various tampering conditions:

- **Rotation attacks**: Images were rotated by angles ranging from $-15°$ to $+15°$ to test the model's ability to handle orientation variations.
- **Scaling attacks**: Images were resized with scaling factors between 0.8 and 1.2, assessing detection performance across size changes.

- **Noise injection**: Two types of noise were added — Gaussian noise with a mean of 0 and variance of 0.01, and salt-and-pepper noise with a density of 0.05 — to simulate pixel-level disturbances.

## A.8 INFERENCE TIME MEASUREMENT

To evaluate the real-time detection capability of our proposed method, we measured the inference time for processing a single 512×512 CT image. The experiments were conducted on a dual NVIDIA RTX 4090 GPU setup, and the average processing time was approximately 0.28 seconds per image. This demonstrates that our framework is capable of near-real-time detection, making it suitable for practical applications in clinical settings. Further optimizations, such as model pruning and hardware acceleration, could be explored in future work to enhance computational efficiency and reduce detection latency.

## A.9 VISUAL COMPARISON SAMPLES

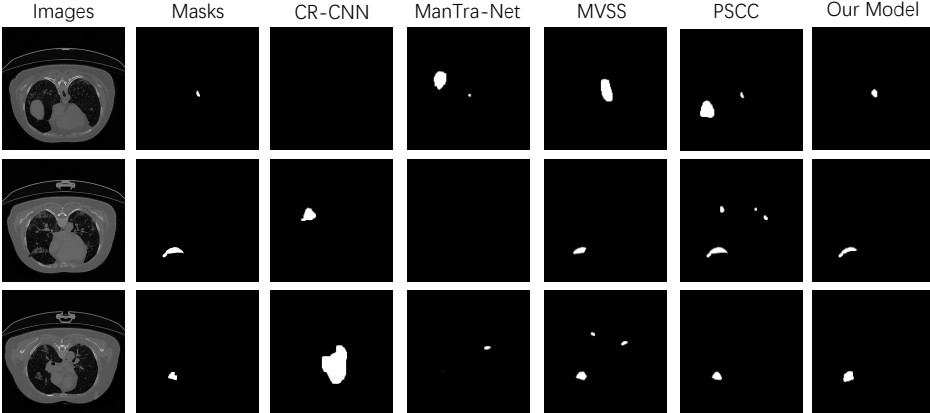

Figure 4: Examples of Image Tampering Detection with State-of-the-art Methods.

Figure 4 presents qualitative comparisons between our proposed method and several state-of-the-art image tampering detection approaches on representative samples from different datasets. In each example, the left column shows the original input image, followed by the ground truth tampered regions, and the detection results produced by competing methods, including CR-CNN, ManTra-Net, HiFi-Net, CAT-Net, MVSS, PSCC, and our model. It can be observed that our method achieves more precise localization of tampered areas, with clearer boundaries and fewer false detections, particularly in complex scenarios involving subtle noise artifacts or irregular edges.

