# OpenReview forum: "A Dual-Branch Feature Fusion Framework based on GenAI for Robust Detection of Medical CT Image Tampering"
_ICLR.cc/2026/Conference — ICLR 2026 Conference Desk Rejected Submission_

### Official Review · Reviewer_LdsV · 2025-10-24

**Soundness:** 2
**Presentation:** 2
**Contribution:** 2
**Rating:** 2
**Confidence:** 3

**Summary:**

This paper proposes a dual-branch feature fusion framework based on GenAI for detecting tampering in medical CT images.
The main idea is to use a generator to synthesize tampered or forged CT images that help train a dual-branch discriminator, which separately models edge and noise cues and fuses them via a cross self-attention and adaptive feature pyramid (AC-FPN) module.
The authors also introduce a CT tampering dataset (6,000 images, including copy-move and splicing manipulations) and conduct experiments on both public datasets (CASIA, Columbia, NIST16, MSMC) and the proposed CT dataset, showing improved detection accuracy and robustness against perturbations such as rotation and noise.

**Strengths:**

1.Addresses an important real-world problem — tampering in medical CT imaging — with potential relevance for medical forensics and data integrity.

2.Proposes a clear modular structure (edge and noise branches with fusion), offering some interpretability for feature attribution.

3.Introduces a CT tampering dataset and performs evaluations on multiple public datasets.

4.Includes basic robustness and efficiency analyses (rotation, noise, inference time).

**Weaknesses:**

1.Limited novelty – The framework is an incremental combination of known techniques (GANs, SRM filters, U-Net, FPN, attention). There is no clear theoretical innovation or methodological breakthrough suitable for ICLR.

2.Insufficient Figure Captions: A significant weakness is the lack of necessary explanation in the image captions. Figure captions must be self-contained and clearly articulate what the figure shows and how it relates to the paper's claims. The current state forces the reader to search the main text for critical context, diminishing the quality of presentation.

3.Lack of comparative fairness – It is unclear whether baseline models were re-trained under consistent settings or if results were borrowed from prior works.

4.Insufficient Transparency of Key Components: The acquisition or generation process of the anatomical prior map used within the AC-FPN module is not clearly described. This is a crucial architectural element, and the lack of clarity severely impacts the reproducibility of the proposed method.

**Questions:**

1.Were baseline models re-trained under the same data splits and preprocessing pipelines?

2.Formatting, Figure Redrawing, and Captions : The current single-column layout exhibits serious formatting issues, which significantly hinder readability. For example, in the vicinity of Figure 3, the text is severely truncated, resulting in lines containing only two or three words where the remaining space is occupied by the image. Furthermore, figure captions lack necessary explanatory detail, making the diagrams difficult to interpret without extensive reference to the main text. Please strictly adhere to standard ICLR formatting guidelines, ensuring a clean and continuous text flow, and professionally redraw all figures with comprehensive, self-contained captions.

3.Detailed Explanation of "Anatomical Prior Map": Please clearly explain the source and specific generation process of the anatomical prior map used in the AC-FPN module. Is it a pre-computed anatomical segmentation mask, and what is its quantitative impact on the final result?

---

### Official Review · Reviewer_rbN8 · 2025-10-24

**Soundness:** 2
**Presentation:** 2
**Contribution:** 2
**Rating:** 2
**Confidence:** 3

**Summary:**

The paper targets tampering detection in medical CT images and proposes a dual-branch feature fusion framework based on generative AI (GenAI). The method employs a ResNet-style generator during training to produce tampered samples enriched with edge and noise characteristics, and a dual-branch discriminator that learns noise and edge cues separately. On this basis, a fusion module is designed that combines cross self-attention, residual connections, and dynamic feature selection to better model complex cross-branch relationships and adaptively allocate feature weights. The authors also construct a CT tampering dataset and conduct comparative experiments on public tampering datasets and the self-built CT dataset, demonstrating advantages in accuracy and robustness. The overall goal is to use adversarial generation to increase training difficulty and diversity, enhance the detection of subtle artifacts and complex tampering, and maintain generalization.

**Strengths:**

(1) The paper is highly targeted to the medical scenario: it emphasizes explicit modeling of fine-grained artifacts such as subtle noise patterns and edge discontinuities in CT images; the problem motivation is clear and of practical value. The overall framework is logically consistent: the generator increases training sample difficulty, the discriminator adopts a dual-branch divide-and-conquer strategy, and evidence is integrated at the fusion stage; the overall pipeline is clear and engineering-feasible.
(2) The fusion design is targeted: attention and dynamic weighting are used to strengthen cross-branch dependencies and suppress redundancy, which theoretically helps improve separability and robustness. The authors evaluate on multiple public datasets and a self-built dataset, and provide relatively detailed ablation studies to analyze the contributions of key components. Experimental results show that the proposed method outperforms several baseline models on multiple metrics. Meanwhile, building a CT tampering dataset and comparing against public baselines help promote evaluation norms and reproducibility in this subfield.

**Weaknesses:**

(1) Although the GenAI-driven dual-branch feature fusion framework achieves good tampering detection performance on CT images, there are still shortcomings, and the in-depth analysis of the working mechanism is insufficient.
(2) The boundary of innovation is not clearly defined: dual branches (noise/edge) and attention-based fusion have prior work in image forensics/tampering detection. The paper needs to more clearly articulate the essential differences and incremental contributions relative to existing two-stream/multi-branch and GAN-assisted training methods (e.g., explicit modeling of CT-specific noise statistics, formal derivation of the fusion objective). The current presentation appears more like a reasonable module combination rather than a new paradigm.
(3) Insufficient causal evidence for the generator’s contribution: there is a lack of systematic ablations and rigorous controls (with/without generator, different generation objectives and constraints), especially for “unseen tampering types” and “cross-dataset testing,” where the net gain of generation-based enhancement for generalization has not been sufficiently demonstrated.

**Questions:**

(1) The fonts in Figures 1 and 3 are too small to clearly see implementation details; please enlarge the font size and improve layout clarity.
(2) If possible, please add the following experiments:
       1. Systematic comparison with the most related two-stream/multi-branch and GAN-assisted detection methods to clarify the core innovations in the CT-specific scenario, along with corresponding ablations and controls to demonstrate incremental contributions.
       2. Causal evidence of the generator’s contribution: under a unified training budget and data scale, compare performance with/without the generator and with different generation objectives (emphasizing edges/noise); focus on improvements for unseen tampering types and cross-dataset settings, with statistical significance tests.
(3) Figures and formatting: the structural hierarchy of the main model diagram (Figure 1) is not clear; consider re-layout and annotate data flow and branch interfaces. Some tables and figures have small font sizes; unify and improve readability. In the appendix, provide more localization visualizations and failure cases to show the complementarity between the noise and edge branches; also add more visualization experiments to enhance the richness and persuasiveness of the results.

---

### Official Review · Reviewer_8B16 · 2025-10-30

**Soundness:** 2
**Presentation:** 2
**Contribution:** 2
**Rating:** 2
**Confidence:** 5

**Summary:**

This paper proposes a method for detecting tampering in CT images. The paper is well-structured and addresses an interesting problem in the domain of medical image forensics. However, I have several significant concerns regarding the motivation, the novelty of the proposed method, and the experimental design. Given the high standards of ICLR, I am afraid I cannot recommend this paper for acceptance at this time. I believe the work, with substantial revision, might be a better fit for a more specialized, application-focused venue such as MICCAI, ISBI, or BIBM, where the clinical relevance and specific challenges could be further explored.

**Strengths:**

The paper is well-structured and addresses an interesting problem in the domain of medical image forensics.

**Weaknesses:**

1. While the problem of image tampering is important, the motivation for focusing specifically on CT images could be strengthened. The paper would benefit from a more detailed discussion of the threat model. For instance, at what stage of the clinical workflow would such tampering occur (e.g., at the scanner, in the PACS)? What are the concrete real-world risks or documented cases that drive this research? Without a clearer discussion of these points, the significance of the problem remains somewhat speculative and the motivation feels underdeveloped.

2. The authors claim that existing methods "face technical limitations when dealing with complex tampering scenarios, including insufficient ability to recognize different types of tampering." I find this claim to be overstated. The problem of detecting complex tampering scenarios has been considered in prior work (e.g., MIML@CVPR'24), and the challenge of recognizing different tampering types has also been explicitly addressed (e.g., HiFi-Net).

3. The proposed detection network appears to be quite generic and could seemingly be applied to natural images without much modification. The use of standard components like edge enhancement and SRM, which are common in natural image forensics, raises questions about the method's specific adaptation to CT imagery. The paper does not seem to incorporate any prior knowledge of CT imaging physics or specific anatomical constraints to explicitly address the challenges outlined in the introduction (e.g., high diversity, complex scenes).

4. The paper argues that a key difference between CT and natural images is the high diversity of CT scans. This assertion is counter-intuitive. While there is patient variability, the domain of natural images (with countless objects, textures, lighting conditions, and complex scenes) is arguably far broader and more diverse than the relatively constrained domain of anatomical scans from CT devices.

5. The selection of comparative methods is a significant weakness. The baselines are not representative of the current SOTA in image manipulation detection. The most recent method cited, HiFi-Net, was designed for a different task (hierarchical semantic segmentation) and its comparison here is not entirely fair. A more appropriate and recent baseline would be PSCC-Net (2022), and its absence makes it difficult to properly assess the proposed method's performance against the current SOTA.

6. There is a discrepancy between the claims and the experiments. The introduction emphasizes the challenge of handling diverse tampering types, yet the quantitative results in Table 2 only report performance on two types of manipulation. The evaluation should be expanded to encompass the variety of tampering scenarios discussed.

7. The authors report a much larger performance gain on the natural image dataset (NIST) compared to their custom CT dataset. This is a curious result that warrants a detailed analysis, which is currently missing. Does this imply the method is, in fact, better suited for natural images than for CT images? This could undermine the central premise of the paper.

8. The robustness evaluation includes image rotation. This is not a realistic transformation for CT scan data in a clinical context, as axial scans have a fixed anatomical orientation. This experiment seems irrelevant and suggests a lack of domain-specific consideration in the evaluation design.

9. The section on the custom CT dataset generation raises several questions. Are any measures taken to ensure the anatomical plausibility of the manipulated images? The manipulations appear to be limited to lesions. Furthermore, if the tampered lesions are sourced from images acquired by the same CT scanner as the target image, the effectiveness of noise-based forensic methods like SRM, which often rely on detecting inconsistencies in device fingerprints, is questionable. This aspect requires a more thorough justification.

**Questions:**

See Weaknesses.

---

### Official Review · Reviewer_JZv6 · 2025-11-05

**Soundness:** 2
**Presentation:** 3
**Contribution:** 2
**Rating:** 4
**Confidence:** 3

**Summary:**

This paper presents a dual-branch feature fusion framework, leveraging generative AI (GenAI) for robust detection of tampering in medical CT images. The architecture employs a ResNet-based generator to create tampered images enriched with edge and noise features, and a dual-branch discriminator—one branch dedicated to edge details, the other to noise artifacts. Features extracted by the two branches are combined using a cross self-attention mechanism and dynamic feature selection in a feature fusion module. The approach is evaluated using both standard public image tampering datasets and a newly constructed CT tampering dataset, demonstrating promising accuracy and robustness.

**Strengths:**

**Specialization for Medical CT Tampering**: The paper tackles the underexplored area of CT image tamper detection—a domain with unique challenges distinct from general image forensics, such as subtle noise artifacts and anatomical complexity.

**Comprehensive Empirical Evaluation**: The paper benchmarks its framework on a range of datasets, including public (CASIAv2, Columbia, NIST16, and MSMC-16k) and a freshly created CT-specific dataset, with detailed results presented in Tables 1 and 2.

**Dual-branch Feature Extraction**: Methodologically, separating edge and noise features via dual discriminators is a thoughtful design tailored to the unique signatures of tampering in medical images. The architecture, detailed in Figure 1, underscores this intent, showing clearly separated edge and noise analysis streams.

**Weaknesses:**

**Limited Novelty in Core Components**: Much of the architecture—as described in Section 3—reuses established ideas: GANs for augmentation, dual-branch discriminators (common in vision), and attention-based feature fusion. The paper does not convincingly demonstrate architectural novelty, instead combining these previously explored components. References such as PASQUALINO et al. (2024) or CDDFuse (Zhao et al., 2022, see missing related work below), cover very similar territory for dual-branch, multi-feature fusion in medical vision.

**Empirical Evaluation—Narrow Types of Tampering**: The custom CT dataset (Appendix A.5) includes only copy-move and splicing manipulations, omitting clinically important scenarios such as deletion or inpainting. The justification is lack of public data, but this diminishes the generalizability and medical relevance of the proposed method. The omission is clear in both dataset descriptions and Table 2. This flaw restricts applicability in real-world clinical settings, where attackers may use more diverse tampering strategies.


**Positioning Relative to Existing Literature Incomplete**: The most directly relevant baseline—CT-GAN (Mirsky et al., 2019), which precisely targets malicious CT tampering via GANs—is missing entirely from the literature review and comparisons. Similarly, recent domain benchmarks like LuNoTim-CT (Reichman et al., 2024) are omitted, weakening claims of state-of-the-art status and completeness of evaluation.

**Questions:**

**Clarification on Anatomical Prior Maps**: How are anatomical prior maps constructed and integrated in the feature fusion/AC-FPN modules (cf. Figure 2)? Are these hand-crafted, derived from segmentation, or otherwise learned?

**Loss Weighting Sensitivity**: The total loss function employs weights ($\alpha$, $\beta$, $\gamma$, $\delta$) for adversarial, edge, noise, and attention/cross-branch losses. Was an analysis done for sensitivity to these hyperparameters? Are the chosen values robust across datasets?

**Generalization Beyond Two Tampering Modes**: Given the custom CT dataset includes only copy-move and splicing, do you expect your approach to transfer to other types (e.g., inpainting, deletion) without retraining or redesign? Do you have any preliminary results or insights for other clinical tampering methods?

**Potential Overlap with Earlier Dual-branch and Attention-based Methods**: Given prior works (like CDDFuse, Zhao et al. 2022, and CT-GAN), what is the primary conceptual or technical step forward offered by your method? How does your cross-attention fusion differ from, for example, the decomposed fusion strategies in these works?

---

### Note · Program_Chairs · 2026-01-17
**Submission Desk Rejected by Program Chairs**

The following references in this submission do not refer to real documents and/or have major errors in bibliographic information:

 Z Lin, J Wang, and X Tang. Image splicing detection using spectral residual. IEEE Transactions on Information Forensics and Security, 6(3):965-973, 2011.